# Sensor-Based Gait and Balance Assessment in Healthy Adults: Analysis of Short-Term Training and Sensor Placement Effects

**DOI:** 10.3390/s24175598

**Published:** 2024-08-29

**Authors:** Clara Rentz, Vera Kaiser, Naomi Jung, Berwin A. Turlach, Mehran Sahandi Far, Jutta Peterburs, Maik Boltes, Alfons Schnitzler, Katrin Amunts, Juergen Dukart, Martina Minnerop

**Affiliations:** 1Institute of Neuroscience and Medicine (INM-1), Research Centre Jülich, 52425 Jülich, Germany; vera.kaiser@hhu.de (V.K.); naomidj@student.ubc.ca (N.J.); k.amunts@fz-juelich.de (K.A.); m.minnerop@fz-juelich.de (M.M.); 2Heinrich Heine University Düsseldorf, 40225 Düsseldorf, Germany; 3Centre for Applied Statistics, The University of Western Australia, Perth, WA 6000, Australia; berwin.turlach@uwa.edu.au; 4Institute of Neuroscience and Medicine, Brain & Behaviour (INM-7), Research Centre Jülich, 52425 Jülich, Germany; m.sahandi.far@fz-juelich.de (M.S.F.); juergen.dukart@gmail.com (J.D.); 5Institute of Systems Neuroscience, Medical Faculty, Heinrich Heine University Düsseldorf, 40225 Düsseldorf, Germany; 6Faculty of Mathematics and Natural Sciences, Heinrich Heine University Düsseldorf, 40225 Düsseldorf, Germany; jutta.peterburs@medicalschool-hamburg.de; 7Institute of Systems Medicine and Department of Human Medicine, MSH Medical School Hamburg, 20457 Hamburg, Germany; 8Institute for Advanced Simulation (IAS-7), Research Centre Jülich, 52425 Jülich, Germany; m.boltes@fz-juelich.de; 9Department of Neurology, Center for Movement Disorders and Neuromodulation, Medical Faculty, Heinrich Heine University Düsseldorf, 40225 Düsseldorf, Germany; schnitza@med.uni-duesseldorf.de; 10Institute of Clinical Neuroscience and Medical Psychology, Medical Faculty, Heinrich Heine University Düsseldorf, 40225 Düsseldorf, Germany; 11C. and O. Vogt Institute for Brain Research, Medical Faculty, University Hospital Düsseldorf, Heinrich Heine University Düsseldorf, 40225 Düsseldorf, Germany

**Keywords:** gait, balance, training, habituation effects, motion capturing, smartphone, IMU, wearables, sensor placement, home-based

## Abstract

While the analysis of gait and balance can be an important indicator of age- or disease-related changes, it remains unclear if repeated performance of gait and balance tests in healthy adults leads to habituation effects, if short-term gait and balance training can improve gait and balance performance, and whether the placement of wearable sensors influences the measurement accuracy. Healthy adults were assessed before and after performing weekly gait and balance tests over three weeks by using a force plate, motion capturing system and smartphone. The intervention group (n = 25) additionally received a home-based gait and balance training plan. Another sample of healthy adults (n = 32) was assessed once to analyze the impact of sensor placement (lower back vs. lower abdomen) on gait and balance analysis. Both the control and intervention group exhibited improvements in gait/stance. However, the trends over time were similar for both groups, suggesting that targeted training and repeated task performance equally contributed to the improvement of the measured variables. Since no significant differences were found in sensor placement, we suggest that a smartphone used as a wearable sensor could be worn both on the lower abdomen and the lower back in gait and balance analyses.

## 1. Introduction

Gait disorders affect about 32% of the elderly population and are associated not only with lower quality of life and increased fall risk but also with cognitive decline and depression [1,2]. Since gait is an important marker of health, observational gait and stance analysis is common practice in neurology with standardized rating scales for general and disease-specific applications. A clear understanding of a patient’s gait and its change over time is essential for optimized patient care and disease management [3]. Gait analysis is used to monitor the response to a variety of different interventions [4,5]. Although standardized rating scales help to guide clinical assessments by structuring the observations, they may have major drawbacks regarding interrater reliability, sensitivity, and specificity necessary for defining disease state or progression [6,7,8,9,10,11].

Quantitative gait analysis tools offer the ability to detect nuanced changes in gait by measuring spatiotemporal, kinetic, and kinematic gait characteristics [12,13]. These may include stride time, cadence, step width, joint angles, and time spent in each phase of the gait cycle [3]. For example, decreased stride length, decreased cadence, and increased gait variability have been observed in Parkinson’s disease as one of the most frequent neurological movement disorders [12,14].

When evaluating participants over time, and especially when assessing an intervention, it is important to distinguish the effects of the intervention from the habituation effects of performing a test multiple times. While such “practice effects” or habituation effects are well known in neuropsychological tests [15,16], less knowledge is available on motor tasks, and specifically gait and balance. Meyer et al. [17] assessed the habituation effect in patients with multiple sclerosis for various standardized rating scales, including Timed Up and Go test, 2-Minute Walk Test, and Timed 25-Foot Walk Test. After performing these tests on three consecutive days, they found improvements due to habituation effects of up to 50% of the clinically meaningful detectable change and recommended that training tests should be performed before the actual assessment to mitigate the practice effect. Similarly, Keklicek et al. [18] investigated the habituation effect on balance in healthy participants using the Bertec balance test. Participants’ stability improved between the first and second trial and was subsequently maintained (one attempt right after the other). This indicates a stabilization of the habituation effect after the third trial and supports the need for training tests prior to observation in stance tasks. However, Salthouse et al. [19], after analyzing six different cognitive tasks, reported that it might take up to seven years until no retest effect can be found. In a previous feasibility study, we compared a smartphone app to gold standard methods for assessing the effect of a three-week exercise intervention on gait and balance in a sample of healthy adults [20]. Participants improved their motor performance, but our study lacked a comparison to a control cohort not undergoing the exercise program (but performing the weekly gait and balance tests, including three gait tasks and four stance tasks). Here, we now present a group of participants that did not receive the training but completed the gait and balance tests to assess the practice/habituation effect induced only by the gait and balance tests per se and contextualize our previous findings. 

Despite clear advantages, quantitative gait analysis methods are rarely used in clinical practice. This is partly due to the high cost and low ecological validity of the most commonly used systems [21,22,23]. Motion capture systems, while highly accurate, require trained personnel and a controlled laboratory environment [24]. Force plates are expensive and due to their limited length measurements are restricted to short distances and thus limit the ability to study gait variability over long distances [25,26]. Small, portable, and less expensive inertial measurement units (IMUs) address many of the aforementioned issues, but full-body IMU systems require that a sensor is placed on each body segment, which might impede the use of this technology for daily monitoring [25]. Therefore, wearable measurement systems with fewer IMU sensors and the use of IMU-equipped smartphones have recently become of great interest [27,28,29,30,31,32]. IMUs use an accelerometer, a gyroscope, and a magnetometer, which are now standard sensors in modern smartphones [33]. As of 2018, 78% of German adults own a smartphone, making this gait analysis method highly accessible in a person’s daily life [34]. Additionally, smartphone-based measurement systems are inexpensive, allow for the monitoring of everyday gait and postural patterns, and incorporate familiar technology for users. 

In systematic reviews [8,35], pelvis, shank, and feet have been named as the preferred segments for wearable sensor placement. Those studies using the pelvis mostly placed the wearable sensor or smartphone at the lower back (for review, see Hubble et al. [36] or Ilg et al. [31]), since it is close to the center of mass and movement there reflects the movement of the rest of the body [27,30,37]. Likewise, standard gait measurement systems like Mobility Lab [38,39,40] or other motion capturing systems like Xsens (Xsens Technologies B.V., Enschede, The Netherlands, https://www.movella.com/products/motion-capture, accessed on 24 July 2024) use one of their sensors at the lower back by default (L5 region). Differences in sensor placement have been assessed in several studies, but those studies only compared sensor placement on different segments of the body (e.g., pelvis vs. feet), not different placements on one specific segment (e.g., lower back or lower abdomen used as the “pelvis segment”) [41,42,43]. In everyday life or during study participation requiring frequent interaction with the phone, a person may prefer to carry their phone on the front of their body as it is easier to reach and to handle [28]. This is particularly important for older people and patients with movement disorders, whose fine motor skills are often impaired in addition to their gait and already find it difficult to use a telephone. Improving the user-friendliness of a smartphone-based measurement system would facilitate the implementation of the technology in clinical trials with multiple study sites and frequent follow-up visits and would even allow remote recording of gait and stance in patient’s everyday life. Therefore, it is necessary to understand if placing the device at the front or back of the body affects the results of the gait analysis. Even though a few studies used the wearable sensor or smartphone in a waist bag or belt [44], they did not define where the bag was placed [45] or left it open to the participants where to place the waist bag [46]. No comparison to “gold standard placement” at the lower back was performed and no explanation for a deviation from the standard was given. The present study therefore investigates a comparison of gait variables using a smartphone placed on the front vs. on the back of the trunk and compares those to standard laboratory methods using force plates. 

This study builds on previous research by analyzing and distinguishing between training and habituation effects in healthy adults. This distinction is crucial not only for interpreting the outcomes of motor performance assessments but also for applying these findings to patient populations, where distinguishing between genuine intervention effects and practice effects is vital for disease management and the development of intervention strategies. Furthermore, we assessed whether the more convenient front placement of a smartphone sensor has any drawbacks compared to the traditional back placement, which is commonly used in gait analysis. This will help determine whether the more patient-friendly front placement can be used for gait and balance analyses in patient studies without compromising accuracy or whether the back placement is necessary for more precise measurements. If both placements prove equally accurate, the patient-friendly front placement could be recommended for future patient studies as it simplifies the feasibility of gait and balance assessments.

## 2. Materials and Methods

This study consists of three parts: Analysis of habituation effects in healthy adults performing weekly gait and balance tests over three weeks,Re-analysis of previously reported training effects [20] in healthy adults by comparing an intervention group (four times 20 min gait and balance training per week, plus weekly gait and balance tests at home) with a matched control group (only weekly gait and balance tests at home),Comparative analysis of the results of gait and balance analysis obtained with sensor placement on the lower back (gold standard) vs. on lower abdomen (easier positioning into a waist bag). 

### 2.1. Participants

All participants needed to be able to walk safely without a walking aid, have no joint problems (e.g., osteoarthritis, endoprostheses), or other neurological, muscular, or other medical problems affecting gait (e.g., falls, deep brain stimulation).

For the analysis of habituation effects, 26 participants were recruited to perform weekly gait and balance tests. For the re-analysis of training effects, this sample was compared to 25 participants from a previous feasibility study [20], receiving home-based video training in addition to the weekly gait and balance tests. For analyzing the impact of sensor placement on gait and balance analysis, 32 participants were recruited. 

All participants gave their written informed consent in agreement with the Declaration of Helsinki, and the study protocol was approved by the ethics committee of the Faculty of Psychology of the Heinrich Heine University Düsseldorf, Germany (DU01-2021-01).

### 2.2. Tasks and Questionnaires

Participants had two study visits (T1 and T2) in Düsseldorf, Germany. At the study site, participants of the habituation vs. training effects cohort performed gait and balance tests, consisting of three different gait tasks (normal, backward, and tandem gait) and four different stance tasks (narrow, tandem, single leg stance, and narrow stance with eyes closed) on a force plate. Data were additionally measured by a motion capturing system (Mocap System) and individual smartphones of the participants. Participants were instructed to walk at a normal pace for the gait tasks and move as little as possible for the stance tasks. The maximum duration for a stance task was set to 30 s, but time was stopped if the participant held onto something or changed their foot position. During the three-week period at home, both groups repeated the seven gait and balance tests once a week at home. The intervention group additionally performed four trainings per week, 20 min each, containing mobility, strength, and coordination exercises [20]. By questionnaire, the participants’ age, gender, height, weight, and years of education were retrieved, as well as depression and anxiety scores ([47], German version: HADS-D [48]), self-efficacy, optimism and pessimism (SWOP-K9 [49]), and general habitual well-being (FAHW [50]). Their self-efficacy in relation to falls was assessed via the (modified) German version of the Activities-Specific Balance Confidence scale (ABC-D [51]). Scale ranges and definitions can be found in Rentz et al. [20]. At the second study visit, participants again performed the gait and balance tests and completed the FAHW, SWOP-K9 and ABC-D questionnaires.

In the separate sample to test sensor placement, participants had only one study visit and performed a normal walk across the force plate (about 4 m each way) at their normal pace. They performed this task twice for two minutes each, once with the waist bag at the lower back and once with the waist bag in front on the lower abdomen. 

### 2.3. Measurement Systems

A zebris FDM force plate (4.24 m, zebris Medical GmbH, Isny, Germany, https://www.zebris.de/en/medical/stand-analysis-roll-analysis-and-gait-analysis-for-the-practice, accessed on 27 August 2024), an Xsens motion capturing system (Xsens Technologies B.V., Enschede, Netherlands, https://www.xsens.com/motion-capture, accessed on 27 August 2024), and individual smartphones of the participants with the app JTrack Social installed [52] were used to measure gait and stance tasks of the participants. The force plate uses capacitive pressure sensors to capture the pressure distribution during the task. No preprocessing was performed on the force and pressure data, which were recorded with a frequency of 100 Hz. Gait or balance reports were created automatically by the Noraxon myoPressure™ software (Noraxon U.S.A., Inc., Scottsdale, AZ, USA, https://www.noraxon.com/our-products/myopressure/, accessed on 28 August 2024). The Mocap System uses 17 IMUs to record angular velocity, acceleration, atmospheric pressure, and the Earth’s magnetic field. A calibration process (neutral pose, walk, turn, walk back) was performed as described in the MVN User Manual, i.e., to calculate the orientations of the sensors with respect to the corresponding segments. The data were recorded with the Xsens MVN 2020.2 software and stored in the mvnx format after reprocessing in HD. The participants’ smartphones, which recorded accelerometer and gyroscope data, were placed in waist bags at the lower abdomen. Only in the group to test sensor placement was the waist bag placed at the lower back and lower abdomen successively. The gait and balance measurement systems, variable definitions, and feature extraction process have been described in detail by Rentz et al. [20]. It should be noted that the use of various smartphones introduces variability in the gait and balance measurements due to differences in hardware and sensor accuracy.

### 2.4. Statistical Analyses

An a priori calculation for the required sample size (G*Power 3.1.9.7 [53]) indicated a minimum number of 19 participants for one group to determine differences over time (matched pairs). A one-tailed significance level with α = 0.05, 1 − β = 0.95 and a calculated effect size of 0.8 (according to the results of gait velocity and cadence in Miyai et al. [54]) was used. 

Chi-squared tests were used to analyze differences in distribution for gender and handedness. *T*-tests were used to test for differences in age, height, weight, years of education, anxiety and depression scores. A Mann–Whitney test was conducted to compare time spans between the two study visits and the number of missing data collection points at home. 

Paired *t*-tests were conducted for both questionnaire scores and gait and balance variables to allow direct comparison to the previous results of the training group [20], as well as to compare mean differences between sensor placement at the front vs. back. Following the previous publication, Bonferroni correction for multiple comparisons with *p*-values of less than 0.013 (force plate, mocap system) and 0.017 (smartphone) were used for gait, and *p* = 0.025 for balance tasks. For all other statistical analyses, a *p*-value of less than 0.05 was considered significant. Spearman’s rank correlation was calculated to examine the relationship between sensor placement (difference between front and back) and body weight of the participants. The Minimal Detectable Change (MDC) was calculated (MDC_90_ = 1.64 * SEM * √2 and MDC_95_ = 1.96 * SEM * √2) at the 90% and 95% confidence levels (SEM = standard error of measurement). The reliability of measurements was assessed using the intraclass-correlation coefficients (ICCs) with two-way random effects models. 

Boxplots of all gait and balance variables were examined, and extreme outliers, defined as values greater than 3 times the interquartile range above the third quartile, were excluded.

For group comparison, we chose linear models with fixed and random effects as the statistical approach. A linear model using generalized least squares [55,56] was applied to describe three-way interaction effects between questionnaire scores, group (intervention or control), and measurement time (study visit 1, T1, or study visit 2, T2). Four-way interaction effects between gait or balance variable (cadence, stride time, and velocity for gait; sway area, and velocity for balance), group, measurement time, and measurement system used (force plate, Mocap System or smartphone) were investigated by using a linear mixed-effects model [57]. The same model was used separately for the different gait and balance tasks (e.g., backward gait, tandem stance) to decrease model difficulty. Pairwise comparisons were used to display patterns of interaction; however, they may not capture all nuances of the interaction between factors. 

With a generalized least squares fitted linear model, the three-way interactions between the variable, the measurement system, and the placement of the waist bag (front, back) were described. For all linear models, stepwise backward elimination was used to select the most significant predictors and refine the model’s explanatory power.

## 3. Results

### 3.1. Demographic Characteristics

The control group (n = 26) and intervention group (n = 25) did not differ significantly in the distribution of handedness (21 right-handed and 23 right-handed, respectively, *p* = 0.663), and gender (12 female and 13 female, respectively, *p* = 0.891). No statistical differences in age (*p* = 0.253), height (*p* = 0.841), weight (*p* = 0.318), years of education (*p* = 0.345), HADS anxiety score (*p* = 0.264), or depression score (*p* = 0.347) were found (see Table 1 for values). The number of tests actually carried out at home could not be checked. Only the number of smartphone recordings transmitted was available. No difference in the number of missing recordings was found (*p* = 0.331, 0.5 ± 0.8 missing recordings in the control group, 1.0 ± 1.2 in the intervention group, both in the range of 0 to 3). All participants confirmed that they had completed the requested measurements and training completely. Incomplete execution would not have led to any disadvantages regarding reimbursement.

### 3.2. Group Comparison: Questionnaire Scores

No statistically significant three-way interaction effect between questionnaire scores, group, and measurement time was found (*p* = 0.867). After removing the three-way interaction from the model, no significant two-way interactions were found (*p*-values between 0.634 and 0.946). Eventually, in the resulting main-effects only model, neither the effect of the factor “group” (*p* = 0.867) nor of the factor “measurement time” (*p* = 0.398) was significant—only the factor “questionnaire scores” (*p* < 0.0001), indicating that single questionnaire scores differed from each other in general. Additional paired *t*-tests revealed no statistical difference for the single questionnaire scores over time (*p*-values between 0.091 and 0.880; for mean values of both groups, see Table 2). 

### 3.3. Group Comparison: Gait and Balance Scores

To compare the control group without training to the previously published results of the intervention group, paired *t*-tests were conducted. These revealed a significant improvement between study visit 1 and study visit 2 for five variables in the control group: backward gait (BG) stride time (*p* = 0.004) and cadence (*p* = 0.005), as well as tandem gait (TG) stride time, cadence and velocity (all *p* = 0.003), all measured by the force plate. In comparison, the previous results of the intervention group revealed eleven variables with a significant improvement, including measurements from both the force plate and motion capturing system (see Table 3). For example, mean velocity in the normal gait (NG, measured by the force plate) increased non-significantly from 0.96 m/s to 1.00 m/s in the control group (*p* = 0.361) and significantly from 0.98 m/s to 1.09 m/s in the intervention group (*p* = 0.002). On the other hand, mean velocity in tandem gait (measured by the force plate) increased significantly from 0.41 m/s to 0.45 m/s in the control group (*p* = 0.003). The effect did not reach significance in the intervention group with mean velocity increasing from 0.45 m/s to 0.49 m/s (*p* = 0.065). To contextualize these findings, the Minimal Detectable Change (MDC) was calculated and is indicated in Table 3. For example, for mean velocity in the normal gait measured by the force plate, an MDC of 0.26 m/s (at a 95% confidence level, or 0.22 m/s at a 90% confidence level) was found for both groups. This indicates that both changes (0.11 m/s improvement in the intervention group and 0.04 m/s in the control group) did not exceed measurement error and variability. MDC values can be found in Table 3 for variables with significant changes, and all remaining values are shown in Appendix A (Table A1). 

For the stance tasks, a single significant difference (surviving Bonferroni correction) between T1 and T2 was found in the control group: sway velocity in the single leg stance (SS), measured with the smartphone (*p* = 0.013), and one difference for the intervention group: sway velocity in the tandem stance (TS), measured with the force plate (*p* = 0.006).

Secondly, an advanced statistical model (linear model) was used to gain more profound insight into the interaction of the groups, measurement systems, study visits, and variables. Three significant four-way interactions were found, two within the narrow stance (NS, *p*-values 0.016 and 0.044) and one for the narrow stance with eyes closed (NSEc, *p* = 0.003). No statistically significant four-way interaction effect between gait or balance variables, group, measurement time and used measurement system was found for normal gait (NG, *p*-values between 0.620 and 0.973), backward gait (BG, *p*-values between 0.506 and 0.960), tandem gait (TG, *p*-values between 0.054 and 0.822), tandem stance (TS, *p*-values 0.067 and 0.119) and single leg stance (SS, *p*-values 0.215 and 0.848). The interactions are described in detail below. 

For normal gait, stepwise backward elimination terminated at a model that included overall two two-way interactions: one between the variable and measurement time and one between the variable and measurement system used. The individually listed contrasts revealed a difference in step width between force plate and Mocap System (higher step width for force plate, *p* = 0.002) and in step width between T1 and T2 in general, not depending on the group or measurement system (lower step width at T2, *p* = 0.002). 

For backward gait and tandem gait, stepwise backward elimination terminated at a model that included a three-way interaction between the variable, group, and measurement system used (*p*-values < 0.0001 to 0.007), but none of the significant interactions included the measurement time (T1, T2). In backward gait, the individually listed contrasts revealed a difference in the step width between the two groups when measured with the Mocap System (lower step width for intervention group, *p* = 0.0005) and a difference in step width between the force plate and the Mocap System in general (higher step width for the force plate, *p* < 0.0001). In tandem gait, individually listed contrasts showed a difference in step width between the two groups (lower step width for intervention group, *p* < 0.0001, measured with the Mocap System) and a difference in cadence between the two groups (higher cadence in the intervention group, *p* = 0.024, measured with the smartphone), as well as a difference in step width between the force plate and the Mocap System in general (higher step width for the Mocap System, *p* = 0.002 for the intervention group, *p* < 0.0001 for the control group).

For tandem stance and single leg stance, stepwise backward elimination terminated at a model that included a three-way interaction between the variable, group and measurement system used (*p*-values < 0.0001 to 0.002), as well as a three-way interaction between the variable, measurement time and measurement system used in the single leg stance (*p* = 0.043). As for tandem stance, this included a difference in sway area between the intervention and control groups when measured by smartphone (higher sway area in the intervention group) or Mocap System (higher sway area in the control group), as well as a difference between the measurement systems for sway area in both groups (higher sway area for Mocap System compared to smartphone and force plate, and for force plate compared to smartphone); no differences were found for sway velocity. In single leg stance, differences in sway area were found between intervention and control group at T1 and an even greater difference at T2 (higher sway area in the control group, *p* < 0.0001, measured with the Mocap System).

A four-way interaction was present in the narrow stance with eyes closed (see Figure 1 for raw data), indicating a complex interaction of all factors for this task. Individual contrasts showed an increase in sway area between T1 and T2 for the intervention group, measured with the smartphone (*p* = 0.038), and for the control group, measured with the Mocap System (*p* < 0.0001). Another difference was found in sway area between the intervention group and control group at T2, measured with the smartphone (higher sway area in the intervention group, *p* = 0.027) and with the Mocap System (higher sway area in the control group, *p* < 0.0001). Another significant four-way interaction was found in the narrow stance (see Figure 1 for raw data), where individual contrasts revealed a difference in sway area between T1 and T2 for the intervention group (higher sway area at T2, *p* = 0.0002, measured with the smartphone); as well as between intervention group and control group at T1 (higher sway area in the control group, *p* = 0.0003, measured with the Mocap System) and at T2 (higher sway area in the intervention group, *p* = 0.0001, measured with the smartphone; and higher in the control group, *p* < 0.0001, measured with the Mocap System). 

### 3.4. Sensor Placement

In the sensor placement group, 32 middle-aged adults took part in the study (37.1 ± 15.7 years old, 171.5 ± 6.4 cm body height, 72.9 ± 14.9 kg body weight, body mass index 24.7 ± 4.3). 

The interaction model indicated no statistically significant interaction for the three-way interactions of variable, measurement system and sensor placement (*p*-values 0.237 to 0.396). Significant two-way interactions were only found between the variable and measurement system (*p* < 0.005).

Additionally, we checked whether the mean differences between smartphone and force plate values (same task, same variable) were different, depending on the sensor placement (front vs. back). No statistically significant difference was found for any of the variables (*p*-values 0.116 to 0.192, see Table 4).

To test whether the difference between front and back, measured with the smartphone, was dependent on body weight, we calculated the linear correlation between the individual differences and body weights. The weak correlation of 0.21 did not reach statistical significance (*p* = 0.317). 

## 4. Discussion

This study applied three different measurement systems to analyze in healthy adults (a) the impact of habituation effects on an improvement of gait and balance variables before (T1) and after (T2) performing a gait and balance training, and (b) the effect of sensor (smartphone) placement on the body on gait variables in comparison to values obtained with a force plate as the gold standard. 

### 4.1. Training Effect vs. Habituation Effect

Clear and significant training effects are typically anticipated in subjects with age- or disease-related limitations [5,54,58,59,60]. However, healthy control subjects frequently serve as a comparison group in training studies targeting these populations. Therefore, it is crucial to understand whether unintended habituation or training effects in the control group could obscure a potential group-specific effect in the actual target population (patients, older adults) within the study setting.

When using paired *t*-tests within each group to compare mean differences of the first and second study visit, we found significant improvements in both the control group and—as reported earlier—the intervention group [20], mainly in gait and only in two stance variables. Remarkably, there was no overlap regarding the involved variables between both groups. Thus, none of the variables with a significant change at the second study visit in the control group showed significant changes in the intervention group and vice versa. In the control group, participants performed better at follow-up after three weeks in variables related to the more difficult gait tasks (backward and tandem gait), while they performed worse at the second study visit in the single leg stance. Participants of the intervention group also performed better in (a different set of) variables related to backward and tandem gait, but also in normal gait and tandem stance (Table 3). Interestingly, although the significant variables of the control and intervention group did not match, in the backward gait measured with the force plate, all corresponding variables in the respective other group were close to significance (significant before Bonferroni correction, *p*-values between 0.013 and 0.028). This was not the case for normal gait, only for one variable in the tandem gait, and not for the other two measurement systems, indicating a tendency for habituation effects in the backward gait, i.e., participants might show improvements in backward gait after repeating the tasks three times (once weekly) at home. Given that the number of gait and stance tasks as well as the respective variables with significant improvements (across all measurement systems) was higher in our intervention group (eleven improvements in four tasks) compared to our control group (five improvements in two tasks), this could be interpreted cautiously as indicator of (a) a habituation effect for the more difficult tandem gait tasks and especially backward gait, and (b) a training effect of a short-term training of three weeks, which is stronger than the habituation effect, relevant for normal gait and the tandem stance task.

A potential habituation effect indicated by the comparison of the control group and intervention group is partly in line with Meyer et al. [17] and Keklicek et al. [18], who found improvements between the first and second trial in a gait (patients with multiple sclerosis and healthy adults, respectively) or balance task (healthy adults) at much shorter intervals (three repetitions in three days, or seven repetitions in one day, respectively). Probably because of these short intervals, Keklicek et al. [18] found an improvement between the first and second trial but a stabilization afterwards, while our participants had weekly tests and were only assessed in their first and fifth trial (three trials at home), indicating a potential habituation effect after the fifth trial. Of note, in our study, it was not possible to check whether and to what quality the measurements (corresponding to trials 2–4) and training were carried out at home. Our intention was to keep the study setup and instructions for the video-based training and gait and balance tests at home as simple as possible to facilitate the implementation (little technical equipment, clear instructions) and thus maintain the adherence of the participants to the training and reduce the risk of dropouts. Even though this setup limited our ability to comprehensively monitor the correct performance of the training and tests at home, all participants confirmed that training and measurements had been carried out in full, and the number of unsubmitted smartphone data showed no statistical difference between the two groups. However, future studies could explore alternative methods for tracking home-based assessments, such as monitoring the playback time of the training videos or regular phone checks, in addition or instead of more extensive documentation using a camera system, to improve data collection and address this limitation more effectively.

While the studies mentioned above assessed habituation effects, other studies addressed gait and balance training effects in subjects of different age groups or with various diseases using different periods of training and different statistical approaches, mostly repeated measures ANOVA. Applying a three-week balance training program, as in our study, significant improvements in step widths and step width variability were found in healthy older adults [58], or improvements in gait velocity, cadence, and stride length in patients with stroke [59]. Other studies used slightly longer time periods, and Mak et al. [60] recommended a minimum of four weeks of training for gait training and eight weeks for balance training in their review of patients with Parkinson’s disease. A longer time period may lead to higher training effectiveness [61,62], in particular, if exactly those tasks are trained that are also assessed [63,64], but this may also result in a drop in compliance [65]. However, Li et al. [66] compared Tai Chi training to conventional training in their meta-analysis and found that, among the included studies, only Tai Chi training with shorter time periods (<20 weeks) and shorter total duration (<24 h) showed greater improvement compared to conventional training. Chaabene et al. [67] found in their meta-analysis small positive effects of home-based strength and balance training in healthy older adults, but no overall influence of training duration, frequency or session duration. These results demonstrate that there is still no clear consensus on detailed, optimal training designs. 

When considering the calculated MDC in our samples, this indicates that the changes observed within each group applying paired *t*-tests were not large enough to exceed measurement error and variability. 

Since comparing first and second study visit within each group does not allow statistically sound statements regarding group differences, we additionally applied a linear mixed effect model to investigate four-way interaction effects between gait or balance variable, group, measurement time and measurement system. By using this statistically more sophisticated approach, only one interaction hinting towards a potential habituation effect was found: in normal gait step, widths were lower at T2 in general, indicating that participants were able to improve their gait stability between the first and second study visit, independent of group affiliation. With respect to training effects, most gait and balance tasks (five out of seven) revealed no significant different trends over time in our control group vs. intervention group, arguing against a training effect. There are several possible explanations for why both training and habituation effects resulted in significant improvements in gait and balance. The most obvious is that practicing the tasks may have been more challenging than the training itself, leading to improvements through repetition. Alternatively, the initial performance may have been low, so any form of practice—whether through training or habituation—would naturally lead to improvements. Other less likely explanations could involve different underlying mechanisms of improvement, such as increased motivation to perform better with each subsequent attempt. The narrow stance with eyes closed and eyes open were the only two tasks where an interaction between all four factors of interest was found, indicating a group difference over time. However, according to pairwise contrast calculations, these group differences were surprisingly caused by an increase in sway area (worse performance) over time for the intervention group (smartphone) and the control group (Mocap System), arguing against a training effect for these two stance tasks as well. The observed decline was contrary to the literature [68]. Of interest, Uematsu et al. [69] found an effect of a dual-task balance training on dynamic and static balance in a small healthy older adults’ cohort, but similar to our study, their control group without training decreased in their standing time in the single leg stance. We observed a similar decline in performance for two stance tasks in our control and in our intervention group (healthy subjects). Possible reasons could be that by being familiar with the task, participants underestimated it, paid less attention or lost their focus, as it was no longer a challenging new task.

Our approach also revealed some methodological issues. Compared to the control group, participants of the intervention group had lower step widths in backward and tandem gait plus higher cadence in tandem gait as well as lower sway area in narrow stance at both visits, which might hint towards an overall better gait performance of the intervention group, independent of measurement time. Possible reasons for this are unclear since both groups did not differ regarding age or anthropometric data. However, if the intervention group should have had an overall better gait performance, this would have increased the likelihood of depicting significant group differences compared to our control group—which was not the case. This further strengthens our assumption, that there was no relevant training effect in our intervention group. The linear models indicated that step widths measured with the force plate were in general different compared to step widths measured with the Mocap System. These differences are likely attributable to the way these two measurement systems operate—while the former uses pressure capacitive sensors with precise distance information, the latter is prone to imprecision in the calibration process which is used to create an avatar based on biomechanical models. The different measurement systems, mocap system and smartphone, led to contradictory results in three out of four stance tasks (tandem stance, narrow stance with eyes open/closed): using the smartphone, sway area was larger in the intervention group, but using the Mocap System, sway area was larger in the control group. This indicates that, at least for stance tasks, these two measurement systems are not yet highly comparable. Differences in accuracy are known from other studies using smartphones vs. camera-based systems [70,71], or smartphones vs. force plate [72], and should be minimized in the future. In general, the ways of operating and different advantages/disadvantages of the three measurement systems should be considered depending on the study setup and the tasks analyzed: while the force plate, which is considered the gold standard, shows an excellent accuracy but is limited to laboratory environments, Mocap Systems offer a balance between accuracy and flexibility, making them suitable for a broader range of settings, including assessments at home, although with some restrictions regarding space and maximum possible time interval. Smartphones, on the other hand, are less precise but provide the greatest accessibility and ease of use, making them ideal for large-scale studies or at-home monitoring over longer time periods (e.g., to capture real-world data) where cost and convenience are prioritized. Considering that the measuring systems all work in different ways, not only the four-way but also the three-way interactions between gait or balance variable, group and measurement time only would have been of interest for detecting training effects, but no interaction of that kind was found. 

Summing up, both the three-week gait and balance training in the intervention group and just repeating gait and balance tests once per week in the control group led to improvements in the tested gait and stance tasks, but these improvements did not differ and were below measurement error and variability according to the MDC calculation. The additional gait and balance training therefore did not generate a measurable additional benefit in healthy controls. Our future research efforts will focus on examining patient data to build on the insights gained from this study and to explore the implications of our findings in a clinical context. Understanding how certain effects manifest in healthy subjects is crucial for accurately interpreting patient outcomes (particularly when compared with healthy control subjects) and ensuring the relevance of our findings in a clinical setting. Our findings can help to understand how much of the observed improvement might be due to specific training versus general familiarity with the tasks. While we cannot directly infer how these improvements would differ in patients, establishing this baseline in healthy subjects is a critical step toward interpreting studies comparing patient data to a control group more accurately.

### 4.2. Sensor Placement

While previous studies have assessed differences in sensor placement, addressing the placement at different body segments [35,41,42,43,73], our study addresses differences between specific sensor (smartphone) placement at the pelvis (lower back vs. lower abdomen). The lower abdomen might be the most convenient place to put a sensor since, e.g., smartphones used as a sensor device could easily be put into a waist bag or pants pocket [74]. 

Of note, our results did not reveal a statistically significant interaction between the variable, measurement system, and sensor placement, suggesting that the placement of the sensor (front vs. back) does not have an overall influence on the outcome variable. The only significant two-way interaction found was an interaction between the measurement system and variable, indicating that, not surprisingly, the measurement system used (force plate, Mocap System or smartphone) did have an influence on the value of the outcome variable. Any interaction including sensor placement did not reach statistical significance (*p*-values between 0.145 and 0.396). Additionally, calculated differences between the values for each variable obtained with the force plate (gold standard measurement system) and the smartphone did not differ between front vs. back placement of the smartphone. Furthermore, the differences between front and back placement were not dependent on body weight. While the effect of obesity on gait kinematics in general (e.g., shorter step length, longer stance phase) is well known [75], only few studies assessed the influence of BMI on gait analyses with wearable sensors [76,77], but they did not compare their results to a gait analysis system that did not use body-worn sensors. Our sample included healthy adults with on average normal body weight but included both an underweight adult (n = 1, BMI < 18.5) and adults with obesity (n = 3, BMI of 30 or greater), suggesting the generalizability of these findings to a diverse and representative population. 

While previous studies have used wearable sensors mostly at the lower back [31,36], the results of the present study support the use of wearable sensors in a manner (front of body) that is most convenient for participants, which could ideally lead to greater study adherence and lower drop-out rates. Additionally, this increases comparability to other studies which might have used a sensor placement at the lower back. Nevertheless, further studies should address whether our results obtained in a sample of healthy adults with a broad age and weight range can be generalized to patient samples or samples with a more restricted age range.

## 5. Conclusions

In our previous analysis, which was restricted to the intervention group [20], a three-week gait and balance training was able to induce small changes in a group of healthy adults. The here performed comparison with a control group addressed habituation effects and applied more advanced statistical models to compare both groups. Both the training and habituation effects resulted in significant improvements in gait and balance after a three-week period of either weekly tests alone or a combination of training and weekly tests. However, these improvements remained below the calculated measurement error and variability. While three weeks of gait and balance training in healthy adults did not significantly enhance gait and stance patterns beyond the small improvements already achieved through weekly tests alone, the overall positive impact on motor function is promising. This indicates that even minimal interventions could possibly lead to detectable changes in gait and balance in healthy adults and that more intensive training may be necessary to produce distinct training effects. These insights are important for contextualizing patient data, as they provide a baseline for understanding how different types of interventions might influence outcomes in a clinical setting. Based on the results and limitations of the current study, we recommend for future investigations—addressing video-based training at home—to implement monitoring of tasks performed at home (e.g., tracking, regular calls) and at least one baseline measure to reduce habituation effects. In addition, our analysis of the sensor placement (lower abdomen vs. lower back) showed comparable values in healthy adults, which leads to the conclusion that a smartphone as a wearable sensor could also be worn in the position on the lower abdomen, which is probably more comfortable and easier to access for study participants.

Future studies should explore the effects of varying training frequencies, durations and intensities to clarify when the training and habituation effects stabilize and to gain a better understanding of how these factors and effects influence outcomes. 

## Figures and Tables

**Figure 1 sensors-24-05598-f001:**
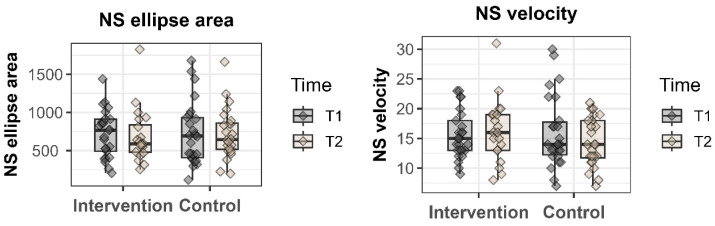
Narrow stance with eyes open (NS) and with eyes closed (NSEc) variables, as the tasks showing the only four-way interaction in the linear models, displayed for the control group and intervention group at both study visits, measured with the force plate.

**Table 1 sensors-24-05598-t001:** Demographic and anthropometric data for control and intervention groups.

Variable	Mean ± SD (Range Min.–Max.)	
	Control (n = 26)	Intervention (n = 25)
Age [years]	38.3 ± 17.4 (21–75)	44.1 ± 18.4 (20–71)
Body Height [cm]	172.8 ± 8.8 (155–185)	172.3 ± 9.9 (154–193)
Body Weight [kg]	72.32 ± 10.64 (57–90) ^2^	67.83 ± 13.83 (43–97) ^1^
Education [years]	14.5 ± 1.9 (12–20)	15.2 ± 3.2 (10–25)
HADS Anxiety [score]	4.3 ± 3.4 (0–14)	3.3 ± 2.8 (0–9)
HADS Depression [score]	2.0 ± 2.2 (0–7)	2.6 ± 2.6 (0–10)

^1^ n = 18; ^2^ n = 19. HADS = Hospital Anxiety and Depression Scale; SD = standard deviation.

**Table 2 sensors-24-05598-t002:** Questionnaire scores at first and second study visit for both groups.

Questionnaire [Score]	Control Group		Intervention Group	
	Mean ± SD (Min.–Max.)		Mean ± SD (Min.–Max.)	
	T1 (n = 26)	T2 (n = 23)	T1 (n = 25)	T2 (n = 21)
SWOP-SE	3.02 ± 0.53 (1.2–4.0)	3.03 ± 0.54 (2.0–4.0)	3.08 ± 0.49 (2.0–3.8)	3.23 ± 0.45 (2.2–4.0)
SWOP-OP	3.23 ± 0.71 (2.0–4.0)	3.33 ± 0.68 (2.0–4.0)	3.24 ± 0.63 (2.0–4.0)	3.12 ± 0.79 (1.5–4.0)
SWOP-PS	1.63 ± 0.61 (1.0–3.0)	1.67 ± 0.65 (1.0–3.0)	1.74 ± 0.61 (1.0–3.0)	1.67 ± 0.71 (1.0–3.0)
ABC-D	18.1 ± 3.11 ^1^ (16–28)	18.7 ± 4.85 (16–33)	18.0 ± 2.57 (16–28)	17.8 ± 2.34 (16–24)
FAHW	56.5 ± 20.7 ^1^ (1–89)	56.3 ± 23.1 (8–86)	59.1 ± 16.8 (21–83)	54.6 ± 25.3 ^2^ (−5–86)

^1^ n = 25; ^2^ n = 20. ABC-D = Activities-Specific Balance Confidence scale, German version; FAHW = questionnaire for general habitual well-being; SD = standard deviation; SWOP = questionnaire for self-efficacy (SE), optimism (OP) and pessimism (PS); T1 = study visit one; T2 = study visit two.

**Table 3 sensors-24-05598-t003:** Side-by-side presentation of intervention group and control group at the first and second study visit.

Measurement System	Variable	Control Group				Intervention Group			
		Mean ± SD (Min.–Max.)		Sign./ MDC95		Mean ± SD (Min.–Max.)		Sign./ MDC95	
		T1	T2		ICC	T1	T2		ICC
Force Plate	NG stride time	1.20 ± 0.10 s	1.16 ± 0.07 s	*p* = 0.053	0.52	**1.20 ± 0.13 s**	**1.13 ± 0.10 s**	***p* = 0.003**	0.71
		(1.02–1.40)	(1.01–1.31)	0.17		**(0.97–1.55)**	**(0.91–1.29)**	0.18	
	NG cadence	1.69 ± 0.14 s^−1^	1.73 ± 0.10 s^−1^	*p* = 0.077	0.53	**1.70 ± 0.17 s^−1^**	**1.80 ± 0.18 s^−1^**	***p* = 0.002**	0.80
		(1.43–1.95)	(1.53–1.98)	0.23		**(1.30–2.08)**	**(1.55–2.20)**	0.22	
	NG velocity	0.96 ± 0.16 m/s	1.00 ± 0.10 m/s	*p* = 0.361	0.51	**0.98 ± 0.14 m/s**	**1.09 ± 0.12 m/s**	***p* = 0.002**	0.59
		(0.58–1.28)	(0.81–1.19)	0.26		**(0.64–1.28)**	**(0.92–1.42)**	0.26	
	NG step width	11.1 ± 2.1 cm	10.6 ± 2.5 cm	*p* = 0.266	0.66	**11.6 ± 2.6 cm**	**10.7 ± 2.5 cm**	***p* = 0.004**	0.91
		(8–16)	(6–16)	3.68		**(7–16)**	**(8–15)**	2.18	
	BG stride time	**1.24 ± 0.13 s**	**1.17 ± 0.12 s**	***p* = 0.004**	0.60	*1.22 ± 0.13 s*	*1.18 ± 0.12 s*	*p = 0.027*	0.68
		**(0.99–1.64)**	**(0.98–1.38)**	0.22		*(1.04–1.56)*	*(0.94–1.37)*	0.20	
	BG cadence	**1.64 ± 0.16 s^−1^**	**1.73 ± 0.17 s^−1^**	***p* = 0.005**	0.61	*1.66 ± 0.16 s^−1^*	*1.72 ± 0.18 s^−1^*	*p = 0.028*	0.68
		**(1.23–2.02)**	**(1.45–2.03)**	0.29		*(1.32–1.92)*	*(1.47–2.12)*	0.27	
	BG velocity	*0.63 ± 0.12 m/s*	*0.70 ± 0.11 m/s*	*p = 0.013*	0.55	**0.69 ± 0.09 m/s**	**0.76 ± 0.09 m/s**	***p* = 0.005**	0.65
		*(0.33–0.78)*	*(0.50–0.89)*	0.22		**(0.53–0.86)**	**(0.61–0.92)**	0.16	
	TG stride time	**1.83 ± 0.34 s**	**1.69 ± 0.31 s**	***p* = 0.003**	0.72	1.66 ± 0.31 s	1.61 ± 0.35 s	*p* = 0.180	0.49
		**(1.33–2.57)**	**(1.25–2.43)**	0.49		(1.19–2.44)	(1.00–2.44)	0.65	
	TG cadence	**1.14 ± 0.22 s^−1^**	**1.24 ± 0.22 s^−1^**	***p* = 0.003**	0.76	*1.23 ± 0.24 s^−1^*	*1.33 ± 0.26 s^−1^*	*p = 0.019*	0.79
		**(0.78–1.52)**	**(0.83–1.62)**	0.30		*(0.68–1.68)*	*(0.85–2.02)*	0.32	
	TG velocity	**0.41 ± 0.09 m/s**	**0.45 ± 0.10 m/s**	***p* = 0.003**	0.67	0.45 ± 0.12 m/s	0.49 ± 0.13 m/s	*p* = 0.065	0.84
		**(0.25–0.58)**	**(0.28–0.64)**	0.15		(0.22–0.72)	(0.25–0.83)	0.14	
	TS sway velocity	43.3 ± 17.2 mm/s	41.7 ± 20.9 mm/s	*p* = 0.419	0.77	**52.3 ± 17.9** **mm/s**	**45.7 ± 22.7 mm/s**	***p* = 0.006**	0.74
		(26–99)	(13–103)	24.92		**(28–107)**	**(22–113)**	28.64	
Mocap System	NG stride time	1.18 ± 0.13 s	1.15 ± 0.10 s	*p* = 0.104	0.59	**1.18 ± 0.13 s**	**1.11 ± 0.10 s**	***p =* 0.002**	0.67
		(0.99–1.36)	(0.98–1.27)	0.15		**(0.94–1.51)**	**(0.93–1.28)**	0.19	
	NG cadence	1.70 ± 0.14 s^−1^	1.75 ± 0.11 s^−1^	*p* = 0.137	0.62	**1.71 ± 0.18 s^−1^**	**1.82 ± 0.17 s^−1^**	***p* = 0.001**	0.77
		(1.47–2.02)	(1.58–2.05)	0.22		**(1.33–2.13)**	**(1.56–2.14)**	0.24	
	BG velocity	0.63 ± 0.12 m/s	0.64 ± 0.14 m/s	*p* = 0.131	0.67	**0.66 ± 0.12 m/s**	**0.75 ± 0.10 m/s**	***p* = 0.007**	0.34
		(0.34–0.78)	(0.25–0.89)	0.21		**(0.31–0.84)**	**(0.58–0.89)**	0.26	
	TG stride time	1.93 ± 0.59 s	1.73 ± 0.37 s	*p* = 0.069	0.75	**1.76 ± 0.42 s**	**1.49 ± 0.23 s**	***p* = 0.003**	0.61
		(1.24–3.76)	(1.20–2.51)	0.69		**(1.17–3.11)**	**(1.00–1.96)**	0.64	
	TG cadence	1.12 ± 0.28 s^−1^	1.21 ± 0.24 s^−1^	*p* = 0.106	0.83	**1.19 ± 0.25 s^−1^**	**1.35 ± 0.18 s^−1^**	***p* = 0.001**	0.65
		(0.53–1.61)	(0.80–1.66)	0.31		**(0.64–1.70)**	**(1.02–1.69)**	0.38	
Smart-phone	SS sway velocity	**30.5 ± 8.9 mm/s**	**44.6 ± 4.4 mm/s**	***p* = 0.013**	0.36	16.9 ± 8.9 mm/s	17.8 ± 4.4 mm/s	*p* = 0.439	0.62
		**(10.9–57.4)**	**(18.5–107.5)**	46.7		(0.9–32.6)	(8.1–22.9)	12.4	

*p*-values highlighted in bold represent significant values after Bonferroni correction (<0.013 for gait tasks, <0.025 for balance tasks), *p*-values in italics represent significant values before Bonferroni correction (<0.05). The Minimal Detectable Change (MDC) for a 95% confidence interval is indicated for both the intervention and control groups. BG = backward gait, ICC = Intraclass Correlation Coefficient, Interv. = intervention group, NG = normal gait, Sign. = significance, SS = single leg stance, TG = tandem gait, TS = tandem stance, T1 = study visit one, T2 = study visit two.

**Table 4 sensors-24-05598-t004:** Comparison of normal gait variables for sensor placement at the front or back.

Variable	Smartphone		Force Plate		
	Mean ± SD Front (Min.–Max.)	Mean ± SD Back (Min.–Max.)	Mean ± SD Front (Min.–Max.)	Mean ± SD Back (Min.–Max.)	Sign.
velocity	0.66 ± 0.17 m/s	0.63 ± 0.15 m/s	1.05 ± 0.15 m/s	1.07 ± 0.14 m/s	*p* = 0.118
	(0.35–1.06)	(0.37–1.10)	(0.83–1.42)	(0.83–1.36)	
str. time	1.31 ± 0.24 s	1.45 ± 0.64 s	1.20 ± 0.10 s	1.19 ± 0.10 s	*p* = 0.192
	(1.03–2.09)	(0.53–3.38)	(1.03–1.37)	(1.00–1.42)	
cadence	1.56 ± 0.26 s^−1^	1.51 ± 0.41 s^−1^	1.68 ± 0.14 s^−1^	1.70 ± 0.14 s^−1^	*p* = 0.116
	(0.93–2.00)	(0.48–1.99)	(1.47–1.95)	(1.42–2.00)	

*p*-values refer to the mean difference between front and back placement, comparing the difference of differences between the two measurement systems. Str. = stride. Differences between the two measurement systems are likely attributable to the way they operate (accelerometry vs. capacitive sensors).

## Data Availability

The sensor data presented in this study are openly available in the pedestrian dynamics data archive at https://doi.org/10.34735/ped.2022.7.

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
