# Peer review of "Sensor-Based Gait and Balance Assessment in Healthy Adults: Analysis of Short-Term Training and Sensor Placement Effects"

_sensors, 2024, doi:10.3390/s24175598_

Round 1

Reviewer 1 Report

Comments and Suggestions for Authors

This paper examines the effects of repeated testing, short-term training and sensor placement on gait and balance tests conducted on healthy adults. The authors conclude that habituation induced by performing weekly gait/balance tests over a period of 3 weeks can improve gait and balance scores, with the effects however remaining below the measurement error. The effects remain the same even if the subjects receive home-based video training in addition to the weekly tests. On the other hand, the placement of the sensor on the front or the back of the body seems to have no effect on the measurement accuracy.

It should be noted that this paper complements a previous publication by the same authors in which they assessed the effect of a 3-week training program on the gait and balance of healthy adults. The previous publication lacked an untrained control sample to decouple training and habituation effects. The control sample is included in this paper, and the effects of training were determined by comparing the results between the two publications.

In my opinion the paper is well written and useful for people who conduct gait and balance tests using wearable devices. The sample size is large (26 and 32 people for the two studies). The literature review included in the paper is very good. The paper examines effects (habituation, training, sensor placement) that are important for the proper design of experiments concerning wearable devices that improve gait and balance. However, I must admit I was hopeful that the paper would conclude by providing evidence for an optimized experimental protocol that would serve as a future reference. Nevertheless, I think the paper is useful and should be published.

Comments:

·        It would have been better perhaps to make the participants record themselves on camera when doing the tests at home. Not knowing whether or how the home tests were conducted is a point of uncertainty.

·        Do the authors have an idea of when the training and habituation effect stabilize (reach saturation)? This would be an interesting study.

·        Can the authors comment: When studying the performing gait/balance measurements, would it be better to use untrained subjects, or fully trained subjects that have reached the saturation stage of the practice effect?

·        The abstract is a bit misleading since it mentions that 51 people were tested. However, the “2.1 Participants” section mentions that 26 people were tested the habituation study, as well as the control sample for the training, and 32 people were tested for the sensor placement study. These 26 people were then compared with the 25 people from the earlier publication to deduce the effects of the training, thus leading to the number 51. I think it should be made clearer in the abstract how many people were tested in this study.

·        I think this study would have been more complete if patients had been included, instead of just testing on healthy individuals. This is because the findings of the study on healthy individuals do not necessarily translate well to patients.

Author Response

We would like to thank the reviewers for their suggestions and for the opportunity to revise our manuscript. Answers to each of the points raised by the reviewers are given below.

We have now added a new section to the Conclusion, which includes recommendations for future research including an optimized experimental protocol and a clearer statement on the significance of the findings from the current study.

Comment 1: It would have been better perhaps to make the participants record themselves on camera when doing the tests at home. Not knowing whether or how the home tests were conducted is a point of uncertainty.

Answer 1: We appreciate the Reviewer's suggestion and agree that this information would be indeed very valuable. With our study design we have tried to minimize the effort for the study participants (especially for carrying out the training at home) for various reasons: Firstly, to maximize numbers of participants, as we expected a high drop-out rate if the tasks at home were too complex or time-consuming. Additionally, more equipment would have increased study costs, potentially further reducing the number of participants. Secondly, we wanted to keep the study instructions simple, particularly with regard to future patient studies, as we anticipate that patients will have greater challenges with technical set-ups and managing multiple devices. Nevertheless, we recognize the importance of this point and are exploring alternative methods to monitor at-home testing and training in future studies, such as tracking video playback time or weekly phone calls. We have added a sentence in the Discussion (page 12, line 452-463) and in the Conclusion (page 15, line 612-615) to address these considerations:

Our intention was to keep the study setup and instructions for the video-based training and gait and balance tests at home as simple as possible to facilitate the implementation (little technical equipment, clear instructions) and thus maintain the adherence of the participants to the training and reduce the risk of drop-outs. Even though this set-up limited our ability to comprehensively monitor correct performance of the training and tests at home, all participants confirmed that training and measurements had been carried out in full, and the number of unsubmitted smartphone data showed no statistical difference between the two groups. However, future studies could explore alternative methods for tracking home-based assessments, such as monitoring the playback time of the training videos or regular phone checks, in addition or instead of a more extensive documentation using a camera system, to improve data collection and address this limitation more effectively.

Based on the results and limitations of the current study, we recommend for future investigations, addressing video-based training at home, to implement monitoring of tasks performed at home (e.g. tracking, regular calls) and at least one baseline measure to reduce habituation effects.

Comment 2: Do the authors have an idea of when the training and habituation effect stabilize (reach saturation)? This would be an interesting study.

Answer 2: We apologize for not being able to define such an effect based on our data. We considered providing this data based on the weekly tests recorded on the smartphone, but as there was no improvement after intervention/habituation when measured using the smartphone, we are currently unable to define a point of stabilization. Keklicek et al., Meyer et al. and Salthouse et al. summarized that a stabilization occurred between first and third day (walking tasks), between first and second trial (balance tasks) or even after only seven years (cognitive tasks), which is quite inconsistent. We recommend conducting further studies to determine the optimal balance between habituation and training effects, depending on the research question being addressed. Therefore, similar training designs should be investigated across various time spans, training frequencies, and durations. Accordingly, this perspective was included in the Conclusion of the manuscript (page 15, line 612-615 & 620-622):

Based on the results and limitations of the current study, we recommend for future investigations, addressing video-based training at home, to implement monitoring of tasks performed at home (e.g. tracking, regular calls) and at least one baseline measure to reduce habituation effects. (…) Future studies should explore the effects of varying training frequencies, dura-tions and intensities to clarify when the training and habituation effects stabilize and to gain a better understanding of how these factors and effects influence outcomes.

Comment 3: Can the authors comment: When studying the performing gait/balance measurements, would it be better to use untrained subjects, or fully trained subjects that have reached the saturation stage of the practice effect?

Answer 3: Thank you for this very interesting question. In our study, we used untrained subjects, i.e. participants who had no prior practice with the specific gait and balance tasks. Due to conflicting recommendations in the literature (Meyer et al., Keklicek et al., Salthouse et al. [17-19]), we decided to do the measurements without training session to better represent real-life situations, in which people typically have no specific training for such tasks. However, we are aware that using fully trained subjects, who no longer improve their performance by simply repeating the tasks, could also provide important insights, particularly with regard to the effect of more intensive training programs and the long-term stability of gait and balance measurements. We will consider this approach for future studies to see how training might affect the results and how baseline measurement might affect the results. We have added a recommendation in the Conclusion section of the manuscript (page 15, line 612-615 & 620-622):

Based on the results and limitations of the current study, we recommend for future investigations, addressing video-based training at home, to implement monitoring of tasks performed at home (e.g. tracking, regular calls) and at least one baseline measure to reduce habituation effects. (…) Future studies should explore the effects of varying training frequencies, durations and intensities to clarify when the training and habituation effects stabilize and to gain a better understanding of how these factors and effects influence outcomes.

Comment 4: The abstract is a bit misleading since it mentions that 51 people were tested. However, the “2.1 Participants” section mentions that 26 people were tested the habituation study, as well as the control sample for the training, and 32 people were tested for the sensor placement study. These 26 people were then compared with the 25 people from the earlier publication to deduce the effects of the training, thus leading to the number 51. I think it should be made clearer in the abstract how many people were tested in this study.

Answer 4: We agree with the reviewer that the actual number of participants is only clearly stated in the Methods section. We have revised the Abstract to address this recommendation.

Comment 5: I think this study would have been more complete if patients had been included, instead of just testing on healthy individuals. This is because the findings of the study on healthy individuals do not necessarily translate well to patients.

Answer 5: We agree with the Reviewer that the inclusion of patients would provide a more comprehensive understanding, and we plan to include patients in future studies. However, we believe that studying healthy individuals is also essential, as it provides a baseline often used for comparison with patient groups. Our study helps to distinguish between habituation and training effects by showing that both can lead to improvements in healthy subjects. This finding is important to understand the extent to which the observed improvement in patients might be due to specific training or general familiarity with the tasks. While we cannot directly infer from our study how these improvements would differ in patients, establishing this baseline in healthy subjects is a crucial step toward interpreting patient data more accurately. We have added this information to the outlook section of the Discussion (page 14, line 555-564 and page 15, line 610-612):

Our future research efforts will focus on examining patient data to build on the insights gained from this study and to explore the implications of our findings in a clinical context. Understanding how certain effects manifest in healthy subjects is crucial for accurately interpreting patient outcomes (particularly when compared with healthy control subjects) and ensuring the relevance of our findings in a clinical setting. Our finding can help to understand how much of the observed improvement might be due to specific training versus general familiarity with the tasks. While we cannot directly infer how these improvements would differ in patients, establishing this baseline in healthy subjects is a critical step toward interpreting studies comparing patient data to a control group more accurately.

These insights are important for contextualizing patient data, as they provide a baseline for understanding how different types of interventions might influence outcomes in a clinical setting.

Reviewer 2 Report

Comments and Suggestions for Authors

This article makes a valuable contribution in exploring the effects of gait training and sensor position on training effectiveness. The article is clearly structured, the research methodology is appropriate, and the results are comprehensively analyzed. However, there is still room for improvement in some aspects of the article.

Here are some suggestions.

 It is suggested that background information on gait disorders could be more clearly linked to the goals of the current study. Clearly state how the current study fills a gap in the existing literature.

 It is recommended that a more detailed description of how the motion capture system and force plates are calibrated, as well as any potential limitations they may have, especially when compared to smartphone-based measurements, be provided.

 The current interpretation of the discussion on habituation effects seems to focus on the similarities between the intervention and control groups, suggesting a more in-depth analysis of the possible reasons for these effects and how they compare with other studies.

It is recommended that the conclusions be further extended to make clearer recommendations for future research, especially in exploring the long-term effects of training or the potential application of smartphone-based gait analysis in different populations.

Comments on the Quality of English Language

Overall, the paper is fluent, with well-structured sentences that clearly communicate the research ideas and results. There are no obvious grammatical errors, and the articulation of sentences is relatively natural. The use of academic terminology and professional expression is well demonstrated, in line with the norms of academic papers.

Author Response

We would like to thank the reviewers for their suggestions and for the opportunity to revise our manuscript. Answers to each of the points raised by the reviewers are given below.

We have now added a new section to the Conclusion, which includes recommendations for future research including an optimized experimental protocol and a clearer statement on the significance of the findings from the current study.

Comment 1: It is suggested that background information on gait disorders could be more clearly linked to the goals of the current study. Clearly state how the current study fills a gap in the existing literature.

Answer 1: We thank the Reviewer for this helpful suggestion and revised the manuscript accordingly (last paragraph of the Introduction, page 3, line 128-140):

This study builds on previous research by analyzing and distinguishing between training and habituation effects in healthy adults. This distinction is crucial not only for interpreting the outcomes of motor performance assessments but also for applying these findings to patient populations, where distinguishing between genuine intervention effects and practice effects is vital for disease management and the development of intervention strategies. Furthermore, we assessed whether the more convenient front placement of a smartphone sensor has any drawbacks compared to the traditional back placement, which is commonly used in gait analysis. This will help determine whether the more patient-friendly front placement can be used for gait and balance analyses in patient studies without compromising accuracy or whether the back placement is necessary for more precise measurements. If both placements prove equally accurate, the patient-friendly front placement could be recommended for future patient studies, as it simplifies the feasibility of gait and balance assessments.

Comment 2: It is recommended that a more detailed description of how the motion capture system and force plates are calibrated, as well as any potential limitations they may have, especially when compared to smartphone-based measurements, be provided.

Answer 2: Previously, this information was only available in the prior study and referenced in the Methods section of the current study. We agree with the Reviewer that additional details would be beneficial. As a result, we have added a paragraph to both the Methods section (page 4-5, line 197-205) and the Discussion section (page 13, line 536-546), providing more detailed information about the measurement systems and how they compare to each other, including their limitations and advantages/disadvantages:

The force plate uses capacitive pressure sensors to capture the pressure distribution during the task. No preprocessing was performed on the force and pressure data, which were recorded with a frequency of 100 Hz. Gait or balance reports were created automatically by the Noraxon myoPressure™ software. The Mocap System uses 17 IMUs to record angular velocity, acceleration, atmospheric pressure, and the Earth’s magnetic field. A calibration process (neutral pose, walk, turn, walk back) was performed as described in the MVN User Manual, i.e., to calculate the orientations of the sensors with respect to the corresponding segments. The data were recorded with the Xsens MVN 2020.2 software and stored in the mvnx format after reprocessing in HD.

In general, the ways of operating and different advantages/disadvantages of the three measurement systems should be considered depending on the study set-up and the tasks analyzed: While the force plate, which is considered the gold standard, shows an excellent accuracy but is limited to laboratory environments, Mocap Systems offer a balance between accuracy and flexibility, making them suitable for a broader range of settings, including assessments at home, although with some restrictions regarding space and maximum possible time interval. Smartphones on the other hand are less precise, provide the greatest accessibility and ease of use, making them ideal for large-scale studies or at-home monitoring over longer time periods (e.g. to capture real-world data) where cost and convenience are prioritized.

Comment 3: The current interpretation of the discussion on habituation effects seems to focus on the similarities between the intervention and control groups, suggesting a more in-depth analysis of the possible reasons for these effects and how they compare with other studies.

Answer 3: Thank you for your valuable feedback. We now addressed the potential reasons why the group experiencing only habituation or practice effects shows similar improvements to the group receiving targeted training effects (page 12-13, line 494-501). We considered several factors, including the overlap in mechanisms of habituation and training, the impact of repeated exposure and learning curves, and the influence of motivation and attention. These considerations will hopefully help clarify why both groups might show comparable improvements and enhance the discussion of our findings in the context of existing research. We hope we have understood your comments correctly. If not, we would kindly ask you to rephrase your comment, which we will be very happy to address:

There are several possible explanations for why both training and habituation effects resulted in significant improvements in gait and balance. The most obvious is that practicing the tasks may have been more challenging than the training itself, leading to improvements through repetition. Alternatively, the initial performance may have been low, so any form of practice—whether through training or habituation—would naturally lead to improvements. Other, less likely explanations could involve different underlying mechanisms of improvement, such as increased motivation to perform better with each subsequent attempt.

Comment 4: It is recommended that the conclusions be further extended to make clearer recommendations for future research, especially in exploring the long-term effects of training or the potential application of smartphone-based gait analysis in different populations.

Answer 4: We appreciate the Reviewer’s recommendation and agree that it will enhance the study’s clarity and impact. As a result, we have added a new section to the Introduction (page 3, line 128-140), the Discussion (page 14, line 555-564) and the Conclusion (page 15, line 609-615 and line 620-622), which includes recommendations for future research and a clearer statement on the significance of the findings from the current study:

This study builds on previous research by analyzing and distinguishing between training and habituation effects in healthy adults. This distinction is crucial not only for interpreting the outcomes of motor performance assessments but also for applying these findings to patient populations, where distinguishing between genuine intervention effects and practice effects is vital for disease management and the development of intervention strategies. Furthermore, we assessed whether the more convenient front placement of a smartphone sensor has any drawbacks compared to the traditional back placement, which is commonly used in gait analysis. This will help determine whether the more patient-friendly front placement can be used for gait and balance analyses in patient studies without compromising accuracy or whether the back placement is necessary for more precise measurements. If both placements prove equally accurate, the patient-friendly front placement could be recommended for future patient studies, as it simplifies the feasibility of gait and balance assessments.

Our future research efforts will focus on examining patient data to build on the insights gained from this study and to explore the implications of our findings in a clinical context. Understanding how certain effects manifest in healthy subjects is crucial for accurately interpreting patient outcomes (particularly when compared with healthy control subjects) and ensuring the relevance of our findings in a clinical setting. Our finding can help to understand how much of the observed improvement might be due to specific training versus general familiarity with the tasks. While we cannot directly infer how these improvements would differ in patients, establishing this baseline in healthy subjects is a critical step toward interpreting studies comparing patient data to a control group more accurately.

This indicates that even minimal interventions could possibly lead to detectable changes of gait and balance in healthy adults and that more intensive training may be necessary to produce distinct training effects. These insights are important for contextualizing patient data, as they provide a baseline for understanding how different types of interventions might influence outcomes in a clinical setting. Based on the results and limitations of the current study, we recommend for future investigations, addressing video-based training at home, to implement monitoring of tasks performed at home (e.g. tracking, regular calls) and at least one baseline measure to reduce habituation effects. (…) Future studies should explore the effects of varying training frequencies, durations and intensities to clarify when the training and habituation effects stabilize and to gain a better understanding of how these factors and effects influence outcomes.
